# Learning Cuts via Enumeration Oracles

**Daniel Thuerck**
Quantagonia
Bad Homburg, Germany
`daniel.thuerck@quantagonia.com`

**Boro Sofranac**
Quantagonia
Bad Homburg, Germany
`boro.sofranac@quantagonia.com`

**Marc E. Pfetsch**
Department of Mathematics, TU Darmstadt
Darmstadt, Germany
`pfetsch@mathematik.tu-darmstadt.de`

**Sebastian Pokutta**
Zuse Institute Berlin and TU Berlin
Berlin, Germnany
`pokutta@zib.de`

## Abstract

Cutting-planes are one of the most important building blocks for solving large-scale integer programming (IP) problems to (near) optimality. The majority of cutting plane approaches rely on explicit rules to derive valid inequalities that can separate the target point from the feasible set. *Local cuts*, on the other hand, seek to directly derive the facets of the underlying polyhedron and use them as cutting planes. However, current approaches rely on solving Linear Programming (LP) problems in order to derive such a hyperplane. In this paper, we present a novel generic approach for learning the facets of the underlying polyhedron by accessing it implicitly via an enumeration oracle in a reduced dimension. This is achieved by embedding the oracle in a variant of the Frank-Wolfe algorithm which is capable of generating strong cutting planes, effectively turning the enumeration oracle into a separation oracle. We demonstrate the effectiveness of our approach with a case study targeting the multidimensional knapsack problem (MKP).

## 1 Introduction

In this paper, we deal with *integer programs* (IP)

$$\max \{\langle c, x \rangle : Ax \leq b, \ x \in \mathbb{Z}^n\}, \tag{IP}$$

where $A \in \mathbb{Q}^{m \times n}$, $b \in \mathbb{Q}^m$, and $c \in \mathbb{Q}^n$. Let $P := \{x \in \mathbb{R}^n : Ax \leq b\}$ be the underlying polyhedron and its integer hull $P_I := \text{conv}(P \cap \mathbb{Z}^n)$. We restrict attention to the case in which all variables are required to be integral, as the methods we will propose are more readily applicable to this case, but the general idea works for mixed-integer programs (MIP) as well.

Solving IPs is $\mathcal{NP}$-hard in general, however, surprisingly fast algorithms exist in practice [1, 50]. The most successful approach to solving IPs is based on the *branch-and-bound* algorithm and its extensions. This algorithm involves breaking down the original problem into smaller subproblems that are easier to solve through a process known as branching. By repeatedly branching on subproblems, a search tree is obtained. The bounding step involves computing upper bounds for subproblems and pruning suboptimal nodes of the tree in order to avoid enumerating exponentially many subproblems. Upper bounds are generally computed with the help of Linear Programming (LP) relaxations

$$\max \{\langle c, x \rangle : x \in P\}. \tag{LP}$$

Because the integrality constraints are relaxed, optimal solutions of (LP) provide an upper bound for the original problem (IP).

37th Conference on Neural Information Processing Systems (NeurIPS 2023).

Alternatively, *cutting plane* procedures iteratively solve LP-relaxations as long as the solution $x^*$ is not integral (and thus $x^* \notin P_I$). To remove these solutions $x^*$ from the relaxation's polyhedron, one adds cutting planes (or *cuts*) $\langle \alpha, x \rangle \leq \beta$ with $a \in \mathbb{Q}^n$, $\beta \in \mathbb{Q}$, and $\langle \alpha, x^* \rangle > \beta$. The search for such cutting planes or to determine that none exists is called the *separation problem*. The strongest cuts are those that define a *facet*, i.e., the face $P_I \cap \{x : \langle \alpha, x \rangle = \beta\}$ has co-dimension 1 with respect to $P_I$. When the cutting plane method is combined with branch-and-bound, the resulting algorithm is often called *branch-and-cut*. Gomory conducted foundational work in this field, demonstrating that pure cutting plane approaches can solve integer programs with rational data in a finite number of steps without the need for branching [32, 33, 34].

Gomory's initial approach to cutting planes suffered from numerical difficulties at that time, preventing pure cutting plane methods from being effective in practical applications. However, his proposed (Gomory) mixed integer (GMI) cuts are very efficient if combined with branch-and-bound (see the computational study in [11]) and still are one of the most important types of cutting planes used by contemporary solvers. As more GMI cuts are added to a problem, their incremental value tends to diminish. To address this issue, modern MIP and IP solvers use a range of techniques to generate cuts, e.g., mixed-integer-rounding (MIR) inequalities [55], knapsack covers [26, 38], flow covers [39], lift-and-project cuts [5], $\{0, \frac{1}{2}\}$-Chvátal-Gomory cuts [22], and others.

Most cutting plane separation algorithms rely on fixed formulas to derive valid inequalities that separate the target point $x^*$ from the polyhedron $P_I$. An alternative approach is to directly seek to derive the facets of $P_I$ that separate the point $x^*$. Notice that while the facets of the polyhedron $P$ are explicitly known from the problem definition, the facets of its integer hull $P_I$ are unknown in general. While the facet-defining inequalities are intuitively the strongest cuts, they can be relatively expensive to explicitly compute, limiting their applicability in practice. *Local cuts*, a type of cutting planes that try to derive facets of $P_I$, approach this problem by deriving facets of $P_I$ in a reduced dimension, and then *lifting* those cuts to obtain facets in the original dimension. In this paper, we will propose a new variant of the *Frank-Wolfe* algorithm with the goal of *learning* the (unknown) facets of $P_I$ (or at least valid inequalities) in a reduced dimension, which can then be lifted to the original dimension and be used as strong cutting planes. In our learning approach, the underlying polyhedron will only be accessed via an algorithmically simple linear optimization oracle, in contrast to existing approaches, which also need to solve LPs.

## 1.1   Related Work

Local cuts have first been introduced as "Fenchel cuts" in Boyd [13, 14], who developed an algorithm to exactly separate inequalities for the knapsack polytope via the equivalence of separation and optimization. They were subsequently investigated extensively by Applegate et al. [3] for solving the traveling salesman problem (TSP). Buchheim et al. [20, 21] and Althaus et al. [2] adopted local cuts into their approaches for solving constrained quadratic 0-1 optimization problems and Steiner-tree problems, respectively. In [25], Chvátal et al. generalize the local cuts method to general MIP problems.

In the context of knapsack problems, after the aforementioned work of Boyd [13, 14], Boccia [12] introduced an approach based on local cuts, as stated by Kaparis and Letchford [47][1], who further refined the algorithm. Vasilyev presented an alternative approach with application to the generalized assignment problem in [63], see also the comprehensive computational study conducted by Avella et al. in [4]. In [64], Vasilyev et al. propose a new implementation of this approach, with the goal of making it more efficient. In [37], Gu presents an extension of the algorithm of Vasilyev et al. [64].

Existing works on the application of learning methods in solving IP (and more generally MIP) problems can in general be divided into two categories: learning decision strategies within the solvers, and learning heurisitcs to obtain feasible (primal) solutions. Examples of the former would be learning to select branching variables [6, 31, 48], learning to select branching nodes [41], learning to select cutting planes [61], learning to optimize the usage of primal heuristics [23, 42, 49]. A typical example of the latter case would be learning methods to develop *large neighborhood search* (LNS) heuristics [28, 60, 59]. Additionally, a number of works in the literature have focused on learning algorithms for solving specific IP problems [8, 44, 27, 51, 54, 56]. For a more detailed overview of using learning methods in IP, we refer the interested reader to [9].

---

[1] We could not independently verify the claim as we could not access Boccia's paper online.

## 1.2 Contribution

The contributions of this paper can be summarized as follows:

1. We present an efficient, LP-free separation framework that aims to learn local cuts for IPs through the solution of subproblems. We propose to use a variant of the Frank-Wolfe [29] algorithm to solve the associated separation problem. The resulting framework is general and – given the availability of a suitable lifting method – applicable to any IP.

2. We propose a new, dynamic stopping criterion for the application of Frank-Wolfe to the separation problem at hand. This new criterion, derived by exploiting duality information, directly evaluates the strength of the resulting cut and thus dramatically decreases the number of iterations.

3. We illustrate the benefit of our approach in a case study for the multidimensional knapsack (MKP) problem, demonstrating its effectiveness. Our computational results show that embedding our method in the academic solver SCIP leads to 31% faster solving times on the instances solved to optimality, on average.

The rest of this paper is organized as follows: In Section 2, the fundamental framework of local cuts and required notation are introduced. Section 3 presents our approach for the LP-less generation procedure for local cuts. Section 4 demonstrates how the aforementioned framework can be applied to solve the multidimensional knapsack problem. Computational experiments are presented in Section 5. Finally, Section 6 summarizes conclusions and future work.

## 2  Local Cuts

To describe the idea of local cuts, assume that $P \subset \mathbb{R}^n, n > 0$, is a polytope, i.e., bounded, and full-dimensional. Then one considers a small subproblem with underlying polytope $\tilde{P}$, which is usually an orthogonal projection of $P$ onto a lower-dimensional space. The polytope $\tilde{P}$ is restricted to being non-empty and its dimension $0 < k \leq n$ is chosen small enough such that integer optimization problems over $\tilde{P}$ can be solved efficiently in practice, for example, by enumeration. Consider a projection $\tilde{x}$ of the point to be separated $x^*$ on $\mathbb{R}^k$. The procedure tries to generate a valid cut $\langle \tilde{\alpha}, x \rangle \leq \tilde{\beta}$ with $\tilde{\alpha} \in \mathbb{Q}^k$, $\tilde{\beta} \in \mathbb{Q}$, such that $\langle \tilde{\alpha}, \tilde{x} \rangle > \tilde{\beta}$, i.e., it cuts off $\tilde{x}$ from $\tilde{P}$. This cut can be "lifted" to the original space, which yields a cut $\langle \alpha, x \rangle \leq \beta$ that hopefully cuts off $x^*$.[2]

The approach to generate $\langle \tilde{\alpha}, x \rangle \leq \tilde{\beta}$, in the literature mentioned above, relies on the equivalence between optimization and separation [36] and can be very briefly explained as follows. By the Minkowski-Weyl Theorem, we can express $\tilde{P}$ as the convex hull of its vertex set $V$. Let $\tilde{x}^0 \in \tilde{P}$ be an interior point. Then consider the LP

$$\min_{\lambda, \gamma} \{ \gamma : \sum_{v \in V} v\, \lambda_v + (\tilde{x} - \tilde{x}^0)\gamma = \tilde{x}, \ \sum_{v \in V} \lambda_v = 1, \ \lambda \geq 0 \}.$$

The dual problem $(D)$ is

$$\max_{\alpha, \beta} \{ \langle \tilde{x}, \alpha \rangle - \beta : \langle v, \alpha \rangle \leq \beta \ \forall v \in V, \ \langle \tilde{x} - \tilde{x}^0, \alpha \rangle \leq 1 \}.$$

Let $\tilde{\alpha}$, $\tilde{\beta}$ be an optimal solution of $(D)$. Then $\langle \tilde{\alpha}, x \rangle \leq \tilde{\beta}$ is a valid inequality for $\tilde{P}$, since by construction $\langle v, \tilde{\alpha} \rangle \leq \tilde{\beta}$ holds for all $v \in V$ and thus by convexity for all points in $\tilde{P}$. The objective enforces that this cut is maximally violated by $\tilde{x}$ if the optimal value is positive.

Since $\tilde{P}$ may have an exponential number of vertices, problem $(D)$ can be solved by a column generation algorithm (or cutting plane algorithm in the primal). In each iteration, one needs to solve the following pricing problem for the current point $(\hat{\alpha}, \hat{\beta})$: Decide whether there exists $v \in V$ with $\langle v, \hat{\alpha} \rangle > \hat{\beta}$. This can be done by maximizing $\hat{\alpha}$ over the subproblem $\tilde{P}$, i.e., one can use a linear optimization oracle for the subproblem. This subproblem can contain integrality constraints, thereby requiring, again, IP techniques. Note that the most interesting case is where we operate on integer

---

[2]The idea of local cuts is often confused with *lift-and-project* cuts, as the two methods share some high-level ideas, such as exploring solutions in a different space and using projection. However, they are quite different. Lift-and- project methods first lift, then generate a cut and project it back, while local cuts first project and then lift the cut. Moreover, the subproblems to generate cuts are signficiantly different.

hulls, i.e. $\tilde{P} = \tilde{P}_I$ to generate cuts for $P = P_I$. In this way, local cuts can help solving an integer optimization problem over $P$. Hence, in the following sections, any reference to $P$, $\tilde{P}$ holds for the integer case as well and our case study illustrates exactly that.

As mentioned above, the strongest cutting planes are those that define facets. The tilting method by Applegate et al. [3] produces such a facet. Buchheim et al. [20] introduced a different formulation that automatically produces a facet. Chvátal et al. [25] developed a formulation for general MIPs using linear optimization oracles. All three approaches use a sequence LPs at their heart; either for tilting a plane or through a column-generation procedure.

## 3   Learning Strong Cuts from Enumeration

The local cuts framework, applicable to general IPs, relies on a sequence of three operators: SEP, FACET and LIFT. SEP refers to a separation oracle separating the projected point $\tilde{x}$ from $\tilde{P}$ that returns a separating cut $\langle \tilde{\alpha}, x \rangle \leq \tilde{\beta}$ (or certifies that $\tilde{x} \in \tilde{P}$). FACET further refines the cut until it represents a facet of $\tilde{P}$ and lastly, LIFT transforms the resulting facet into the space of $P$ such that it separates $x^*$ from $P$ with high probability. In some variants of local cuts, SEP and FACET may be combined into one step similar to [20], whereas in [25], the *tilting* process is a separate, concrete embodiment of FACET. Note that that facets of the subproblem, when lifted, result in the strongest cuts. In practice, it is often sufficient to find good valid inequalities of $\tilde{P}$. As mentioned before, the original approach for local cuts through duality requires an expensive column-generation method which is based on LPs. In this section, we derive an alternative and LP-less approach.

The general idea of our new approach is sketched in Figure 1: Given a point $\tilde{x} \in \mathbb{R}^n$ that we intend to separate from $P$, we solve the following optimization problem:

$$y^* = \underset{y \in \tilde{P}}{\operatorname{argmin}} f(y), \tag{Separation}$$

with $f(y) := \frac{1}{2}\|y - \tilde{x}\|^2$. Observe that this is effectively the projection of $\tilde{x}$ onto $\tilde{P}$ under the $\ell_2$-norm and that $\nabla f(y) = (y - \tilde{x})$.

We solve (Separation) with a suitable variant of the Frank-Wolfe algorithm. The *Frank-Wolfe algorithm* [29] (also called: *Conditional Gradients* [53]) is a method to minimize a smooth convex function $f$ over a compact convex domain $P$ by only relying on a *First-order Oracle (FO)* for $f$, i.e., given a point $x$ the oracle returns $\nabla f(x)$ (and potentially $f(x)$) as well as a *Linear Minimization Oracle (LMO)* ("oracle" for the remainder of this paper), i.e., given an objective vector $c$, the oracle returns $v \in \operatorname{argmin}_{x \in \tilde{P}} \langle c, x \rangle$. The original Frank-Wolfe algorithm, provided with step sizes $\gamma_t > 0$, iteratively calls the LMO to determine $v_t \leftarrow \operatorname{argmin}_{v \in \mathcal{C}} \langle \nabla f(y_t), v \rangle$ and updates the iterate to $y_{t+1} \leftarrow y_t + \gamma_t(v_t - y_t)$. There are various step-size strategies for $\gamma_t$, but the actual choice is irrelevant for the discussion here; a common choice is $\gamma_t = \frac{2}{t+2}$.

The main advantages of using Frank-Wolfe are (1) if there is a LP-less oracle, valid inequalities can be generated without solving LPs, (2) the computational overhead of the Frank-Wolfe steps compared to calls to the LMO are very light and, finally, (3) as we will show, for the case of (Separation), we can derive a new dynamic stopping criterion that can dramatically reduce the number of iterations. Note that we are not guaranteed to end up with facets, especially when the method is stopped early, however, valid inequalities that are "close" to being a facet can still serve as strong cutting planes.

For our problem minimizing $f$, the Frank-Wolfe algorithm iteratively calls the oracle and updates its current iterate through a convex combination of the previous iterate and oracle's solution vertex. Step by step, the solution is thus expressed through a convex combination of vertices in $\tilde{P}$ as shown in Figures 1a – 1c. At convergence, the hyperplane $\langle \nabla f(y^*), x \rangle \geq \langle \nabla f(y^*), y^* \rangle$ forms the desired cut.

### 3.1   Separation via Conditional Gradients

Let $y^* \in \tilde{P}$ be an optimal solution to (Separation) and let $x \in \tilde{P}$ be arbitrary. By convexity, it follows that $0 \leq f(x) - f(y^*) \leq \langle \nabla f(x), x - y^* \rangle \leq \max_{v \in \tilde{P}} \langle \nabla f(x), x - v \rangle$ and the last quantity is referred to as *Frank-Wolfe gap (at $x$)*. Moreover, the following lemma holds, which is a direct consequence of the first-order optimality condition.

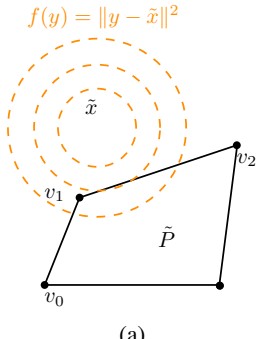
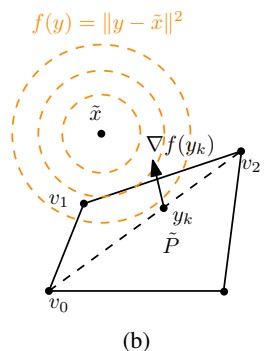
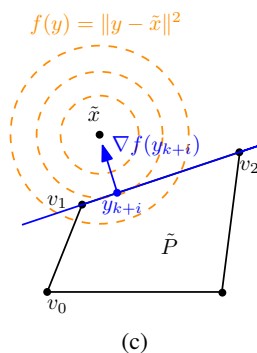

(a)             (b)             (c)

Figure 1: We propose the following approach to separate a fractional point $\tilde{x}$ from a full-dimensional polytope $\tilde{P}$: We solve $\min_{y \in \tilde{P}} f(y) := \frac{1}{2}\|y - \tilde{x}\|^2$, i.e., the $L_2$ projection of $\tilde{x}$ onto $\tilde{P}$, through a variant of the Frank-Wolfe algorithm. Starting from a random vertex (a), the algorithm iteratively computes the gradient of $f$ at the current iterate $y_k$ and uses an *oracle* to solve a linear integer optimization problem over $\tilde{P}$, building up an *active set* of vertices that form iterates through a convex combination (b). At convergence (c) the optimal solution $y^* = y_{k+i}$ together with its gradient forms a cut that induces a facet of $\tilde{P}$: $\nabla f(y_{k+i})^\top x \geq f(y_{k+i})^\top y_{k+i}$ (except for degenerate cases).

**Lemma 1** (First-order Optimality Condition). *Let* $y^* \in \tilde{P}$. *Then* $y^*$ *is an optimal solution to* $\min_{y \in \tilde{P}} f(y)$ *if and only if* $\langle \nabla f(y^*), y^* - v \rangle \leq 0$ *for all* $v \in \tilde{P}$ *(and in particular* $\max_{v \in \tilde{P}} \langle \nabla f(y^*), y^* - v \rangle = 0$*).*

Note that in the constrained case, it does not necessarily hold that $\nabla f(y^*) = 0$, if $y^*$ is an optimal solution. In fact, if the $\tilde{x}$ that we want to separate is not contained in $\tilde{P}$, then $f(y^*) > 0$ and $\nabla f(y^*) \neq 0$ since $y^*$ will lie on the boundary of $\tilde{P}$.

It turns out that we can naturally use an optimal solution $y^* \in \tilde{P}$ to (Separation) to derive a separating hyperplane. By Lemma 1:
$$\langle \nabla f(y^*), y^* \rangle \leq \langle \nabla f(y^*), v \rangle, \tag{Cut}$$
which holds for all $v \in \tilde{P}$. Moreover, if $\tilde{x} \notin \tilde{P}$, then (Cut) is violated by $\tilde{x}$, i.e., $\langle \nabla f(y^*), y^* \rangle > \langle \nabla f(y^*), \tilde{x} \rangle$, since $\langle \nabla f(y^*), y^* - \tilde{x} \rangle \geq f(y^*) - f(\tilde{x}) = f(y^*) > 0$.

Usually, however, we do not solve Problem (Separation) exactly, but rather up to some accuracy. In fact, the Frank-Wolfe algorithm often uses the Frank-Wolfe gap as a stopping criterion, minimizing the function until for some iterate $y_t$ it holds $\max_{v \in \tilde{P}} \langle \nabla f(y_t), y_t - v \rangle \leq \varepsilon$ for some target accuracy $\varepsilon$; note that the Frank-Wolfe gap converges with the same rate (up to small constant factors) as the primal gap (see e.g., [46]). Given an accuracy $\varepsilon > 0$, we obtain the valid inequality
$$\langle \nabla f(y_t), y_t \rangle - \varepsilon \leq \langle \nabla f(y_t), v \rangle, \tag{approxCut}$$
for all $v \in \tilde{P}$, which also separates $\tilde{x}$ from $\tilde{P}$ if it is $\sqrt{\varepsilon}$-far from $\tilde{P}$, i.e., $\|y^* - \tilde{x}\| > \sqrt{\varepsilon}$:
$$\langle \nabla f(y_t), y_t - \tilde{x} \rangle - \varepsilon \geq f(y_t) - f(\tilde{x}) - \varepsilon \geq f(y^*) - \varepsilon > 0.$$
The accuracy $\varepsilon$ is chosen depending on the application; see also [17] for a sensitivity analysis for conditional gradients.

### 3.1.1 A dynamic stopping criterion

It turns out, however, that in our case of interest, the above can be significantly improved by exploiting duality information. This allows us not only to stop the algorithm much earlier, but we also obtain a separating inequality directly from the associated stopping criterion and duality information.

The stopping criterion is derived from a few simple observations, which provide a new characterization of a point $\tilde{x}$ that can be separated from $\tilde{P}$. Our starting point is the following standard expansion. Let $v \in \tilde{P}$ be arbitrary and let $y_t$ be an iterate from above. Then,
$$\|\tilde{x} - v\|^2 = \|\tilde{x} - y_t\|^2 + \|y_t - v\|^2 - 2\langle y_t - \tilde{x}, y_t - v \rangle,$$

which is equivalent to

$$\langle y_t - \tilde{x}, y_t - v \rangle = \tfrac{1}{2}\|\tilde{x} - y_t\|^2 + \tfrac{1}{2}\|y_t - v\|^2 - \tfrac{1}{2}\|\tilde{x} - v\|^2. \tag{1}$$

Observe that the left hand-side is the Frank-Wolfe gap expression at iterate $y_t$ (except for the maximization over $v \in \tilde{P}$) since $\nabla f(y_t) = y_t - \tilde{x}$.

**Necessary Condition.** Let us first assume $\|y_t - v\| < \|\tilde{x} - v\|$ for all vertices $v \in \tilde{P}$ in some iteration $t$. Then (1) yields

$$\langle y_t - \tilde{x}, y_t - v \rangle < \tfrac{1}{2}\|\tilde{x} - y_t\|^2. \tag{altTest}$$

If $v_t$ is the Frank-Wolfe vertex in iteration $t$, we obtain:

$$\begin{aligned}
\tfrac{1}{2}\|y_t - \tilde{x}\|^2 - \tfrac{1}{2}\|y^* - \tilde{x}\|^2 &= f(y_t) - f(y^*) \\
&\leq \max_{v \in P} \langle \nabla f(y_t), y_t - v \rangle = \langle \nabla f(y_t), y_t - v_t \rangle = \langle y_t - \tilde{x}, y_t - v_t \rangle < \tfrac{1}{2}\|\tilde{x} - y_t\|^2.
\end{aligned}$$

Subtracting $\tfrac{1}{2}\|\tilde{x} - y_t\|^2$ on both sides and re-arranging yields: $0 < \tfrac{1}{2}\|y^* - \tilde{x}\|^2$, which proves that $\tilde{x} \notin \tilde{P}$. Moreover, (1) also immediately provides a separating hyperplane: observe that (altTest) is actually a linear inequality in $v$ and it holds for all $v \in \tilde{P}$ since the maximum is achieved at a vertex. However, for the choice $v = \tilde{x}$ the inequality is violated.

**Sufficient Condition.** Suppose that in each iteration $t$ there exists a vertex $\bar{v}_t \in \tilde{P}$ (not to be confused with the Frank-Wolfe vertex), so that $\|y_t - \bar{v}_t\| \geq \|\tilde{x} - \bar{v}_t\|$. In this case (1) ensures:

$$\langle y_t - \tilde{x}, y_t - \bar{v}_t \rangle = \tfrac{1}{2}\|\tilde{x} - y_t\|^2 + \tfrac{1}{2}\|y_t - \bar{v}_t\|^2 - \tfrac{1}{2}\|\tilde{x} - \bar{v}_t\|^2 \geq \tfrac{1}{2}\|\tilde{x} - y_t\|^2.$$

Thus, the Frank-Wolfe gap satisfies in each iteration $t$ that

$$\max_{v \in \tilde{P}} \langle \nabla f(y_t), y_t - v \rangle \geq \langle y_t - \tilde{x}, y_t - \bar{v}_t \rangle \geq \tfrac{1}{2}\|\tilde{x} - y_t\|^2,$$

i.e., the Frank-Wolfe gap upper bounds the distance between the current iterate $y_t$ and point $\tilde{x}$ in each iteration. Now, the Frank-Wolfe gap converges to $0$ as the algorithm progresses, with iterates $y_t \in \tilde{P}$, so that with the usual arguments (compactness and limits etc.) it follows that $\tilde{x} \in \tilde{P}$. In total, we obtain the following result.

*Characterization* 2. The following are equivalent:

1. (Non-Membership) $\tilde{x} \notin \tilde{P}$.

2. (Distance) There exists an iteration $t$, so that $\|y_t - v\| < \|\tilde{x} - v\|$ for all vertices $v \in \tilde{P}$.

3. (FW Gap) For some iteration $t$, $\max_{v \in \tilde{P}} \langle y_t - \tilde{x}, y_t - v \rangle < \tfrac{1}{2}\|\tilde{x} - y_t\|^2$.

In particular, Characterization 2.3 can be easily tested within the algorithm, since the Frank-Wolfe gap is computed anyways. Using this criterion significantly improves the performance of the algorithm. Moreover, the characterization above can also be combined with standard convergence guarantees to estimate the number of iterations required to either certify non-membership or membership (up to an $\varepsilon$-error): If we use the vanilla Frank-Wolfe algorithm, then by standard guarantees (see e.g., [16]) it is known that the Frank-Wolfe gap $g_t = \max_{v \in \tilde{P}}\langle y_t - \tilde{x}, y_t - v \rangle$ satisfies $\min_{0 \leq \tau \leq t} g_\tau \leq \frac{4LD^2}{t+3}$ for appropriate positive constants $L$ and $D$. Suppose that $\max_{v \in \tilde{P}}\langle y_t - \tilde{x}, y_t - v \rangle \geq \tfrac{1}{2}\|\tilde{x} - y_t\|^2$ holds for all iterations $0 \leq t \leq T$. We want to estimate how long this can hold. If $\tilde{x} \notin \tilde{P}$, then using the convergence guarantee yields:

$$0 < \tfrac{1}{2}\operatorname{dist}(\tilde{x}, \tilde{P})^2 \leq \min_{0 \leq \tau \leq t} \tfrac{1}{2}\|\tilde{x} - y_\tau\|^2 \leq \frac{4LD^2}{t+3}.$$

Using $L = 1$ as $f(y) = \tfrac{1}{2}\|\tilde{x} - y\|^2$ and rearranging we obtain

$$t \leq T := \frac{8D^2}{\operatorname{dist}(\tilde{x}, \tilde{P})^2} - 3,$$

i.e., after at most $T$ iterations we have certified that $\tilde{x}$ is not in $\tilde{P}$. Guarantees for more advanced Frank-Wolfe variants can be obtained similarly.

---

**Algorithm 1** Lazy Away-Step Frank-Wolfe [18, 19] with explicit active set and early termination

---

**Input:** Point $y_0 \in \tilde{P}$, function $f(y) = \frac{1}{2}\|y - \tilde{x}\|^2$, step-sizes $\gamma_t > 0$, tolerance $\epsilon > 0$, oracle $\Omega$
**Output:** Valid cut $\langle \tilde{\alpha}, x \rangle \leq \tilde{\beta}$ for $\tilde{P}$ with $\langle \tilde{\alpha}, \tilde{x} \rangle > \tilde{\beta}$ or `false` if $\tilde{x} \in \tilde{P}$.

  1: $v_0 \leftarrow \Omega(\nabla f(y_0))$
  2: $\mathcal{S} \leftarrow \{\{\gamma_0, v_0\}(1 - \gamma_0, y_0)\}$, $\phi = \langle \nabla f(y_0), y_0 - v_0 \rangle$
  3: **for** $t = 0$ **to** $t_{\max}$ **do**
  4:     **if** $\|f(y_t)\| < \epsilon$ **then**
  5:         **return** `false`
  6:     **end if**
  7:     $(\lambda_L, v_L) \leftarrow \min_{(\lambda,v) \in \mathcal{S}} \langle \nabla f(y_t), v \rangle$, $(\lambda_A, v_A) \leftarrow \max_{(\lambda,v) \in \mathcal{S}} \langle \nabla f(y_t), v \rangle$
  8:     **if** $\langle \nabla f(y_t), y_t - v_L \rangle \geq \max\{\langle \nabla f(y_t), v_A - y_t \rangle, \frac{\phi}{2}\}$ **then**
  9:         $v_{t+1} \leftarrow v_L, \gamma_{\max} \leftarrow 1$                                     {lazy step}
10:     **else if** $\langle \nabla f(y_t), v_A - y_t \rangle \leq \max\{\langle \nabla f(y_t), x - v_L \rangle, \frac{\phi}{2}\}$ **then**
11:         $v_{t+1} \leftarrow v_A, \gamma_{\max} \leftarrow \frac{\lambda_A}{1 - \lambda_A}$                               {away step}
12:     **else**
13:         $v_{t+1} \leftarrow \Omega(\nabla f(y_t)), \gamma_{\max} \leftarrow 1$
14:         **if** $\langle \nabla f(y_t), y_t - v_{t+1} \rangle < \frac{\phi}{2}$ **then**
15:             $\phi \leftarrow \min\{\langle \nabla f(y_t), y_t - v_{t+1} \rangle, \frac{\phi}{2}\}, \gamma_{\max} \leftarrow 0$             {dual step}
16:         **end if**
17:     **end if**
18:     **if** $\langle y_t - \tilde{x}, y_t - v_{t+1} \rangle < \frac{1}{2}\|\tilde{x} - y_t\|^2$ **then**
19:         **return** cut $\langle -\nabla f(y_t), x \rangle \leq \langle -\nabla f(y_t), v_{t+1} \rangle$              {see Charac. 2.3}
20:     **end if**
21:     $\alpha \leftarrow \min\{\gamma_t, \gamma_{\max}\}$
22:     $\mathcal{S} \leftarrow \{(\lambda(1 - \alpha), v) : (\lambda, v) \in \mathcal{S}\} \cup \{(\alpha, v_{t+1})\}$
23:     $y_{t+1} \leftarrow \sum_{(\lambda,v) \in \mathcal{S}} \lambda v$
24: **end for**

---

## 3.2 Computational Aspects

A common trait of the local cuts framework is that $\tilde{P}$ is accessed implicitly via an oracle returning vertices. By far the simplest black-box oracle for any bounded IP is *enumeration*, which simply evaluates all possible solutions $x$ and picks the best one. If the IP is unbounded, then pure enumeration does not suffice any more and the oracle needs to take the unboundedness into account. For some problems, we can find *problem-specific* algorithms that only enumerate over feasible solutions or otherwise exploit the structure of the problem at hand to reduce the complexity of enumeration. Examples are the *dynamic programming* approach for knapsack problems, see Section 4.1, or directly enumerating $n!$ possible permutations of $n$ items for the *linear ordering problem* (LOP).

Similarly to the enumeration oracle, the *lifting* routine can also avail of problem-specific structure in some cases. In the case of LOPs, the so-called *trivial lifting* lemma holds, that is, facet-defining inequalities of the LOP polytope in dimension $n$ also define facets in dimension $r > n$ [35], meaning that no lifting is needed at all in this case. For knapsack problems, we can again use a dynamic programming approach, see Section 4.2.

In general, to apply our method to a given class of IP probems, one needs three components: i) a projection $P \to \tilde{P}$. ii) An oracle solving linear optimization problems over $\tilde{P}$ to optimality; in order to be practical, the selected $\tilde{P}$ should be such that the oracle runs reasonably fast. iii) A lifting method to lift cuts from $\tilde{P}$ up to $P$.

In our implementation, we use the *Lazy Away-Step Frank-Wolfe algorithm* of [18, 19], which converges linearly for (Separation). We integrate the novel termination criterion from Characterization 2.3, leading to Algorithm 1. This algorithm should be thought of as a more advanced version of the vanilla Frank-Wolfe algorithm. This variant is motivated by the fact that Frank-Wolfe trends towards sparse solutions and hence the oracle will often return previously-seen vertices. Hence, instead of querying the expensive oracle, one stores all previous vertices in a *active set* whose size is controlled through so-called away steps. It provides superior convergence speed both in iterations and

wall-clock time, exploiting the strong convexity of our objective function of the separation problem; we refer the interested reader to [46, 52, 16] for an overview. Lazification, to be thought of as an advanced caching technique, further reduces the per-iteration cost by reusing previously computed LMO solutions. Lastly, we note that in our setting and case study presented in Section 4, the LMO always returns a vertex. Even though this is not a theoretical requirement for the results presented in this paper, we do not consider the alternative case for brevity.

## 4   Case Study: The Multidimensional Knapsack Problem

The *multidimensional knapsack problem* (MKP) is a well-known problem in combinatorial optimization and is strongly $\mathcal{NP}$-hard. It has been used to address various practical resource allocation problems [30]. The problem involves maximizing the total profits of selected items, taking into account $m$ resource capacity (knapsack) constraints. There are $n$ items that contribute profits given by $c \in \mathbb{Z}^n$. The resource consumption of item $j$ for the $i$th knapsack is given by $a_{ij} \in \mathbb{Z}_+$; this defines a matrix $A = (a_{ij}) \in \mathbb{Z}_+^{m \times n}$. The capacities of the knapsacks are given by $b \in \mathbb{Z}^m$. We define binary variables $x \in \{0, 1\}^n$ such that $x_j$ is equal to 1 if item $j$ is selected and 0 otherwise. Then MKP can be expressed as an IP:

$$\max \{\langle c, x \rangle : Ax \leq b, \ x \in \{0, 1\}^n\}. \tag{MKP}$$

There exists abundant literature on the knapsack problems; we refer the interested reader to the recent survey by Hojny et al. [43].

We will also test our approach on the instances of the *generalized assignment problem* (GAP), see Section 5. GAP is a variant of MKP with applications in scheduling [43]. In addition to the constraints from the MKP problem, it is required that each of the $n$ items be assigned to exactly one knapsack. The interested reader can find a survey, more details, and a comprehensive reference list in [4, 58].

In order for our approach to work, we need to provide two things: the *oracle*, presented in Section 4.1, and the *lifting* routine, presented in Section 4.2, cf. Section 2 and Section 3.

We consider each knapsack problem in turn and try to generate inequalities that are valid for each individual knapsack. This has the advantage that there are practically efficient oracles and more importantly efficient lifting processes. The disadvantage is that the cuts might be weaker, since they are valid for all integer solutions for all knapsack constraint instead of their intersection. An alternative would be to consider optimization oracles for the complete set of knapsack constraints as done by Gu [37]. However, then either lifting becomes more computationally demanding or one cannot use lifting.

### 4.1   The Linear Minimization Oracle

The process begins with a solution $x^\star$ of the LP relaxation of the MKP. We create a reduced knapsack problem of dimension $k \in \mathbb{Z}_+, k \leq n$ by removing variables of each knapsack that have integral values (0/1) in the LP relaxation. The oracles now solve the knapsack problems (KP) for each constraint of the form $\max \{\langle c, x \rangle : \langle w, x \rangle \leq C, \ x \in \{0, 1\}^n\}$, with $c \in \mathbb{R}^k$, $w \in \mathbb{Z}_+^k$, $C \in \mathbb{Z}_+$. In practice, the dimension $k$ is rather small (in our test sets, see Section 5, we observe an average $k$ value of 9.6 with a maximal size of 26), allowing for efficient solution approaches. In our implementation, we use a LMO based on *dynamic programming*. We note that we also apply the preprocessing improvements described by Vasilyev et al. [64], before we run the oracle on the reduced problem.

Dynamic programming, as presented by Bellman in 1957 [7], was one of the earliest exact algorithms for solving KPs. Toth [62] presents additional improvements to the algorithm. More recently, Boyer et al. present massively-parallel implementations running on GPUs [15]. The space and time complexity of the dynamic programming algorithm for KP is $\mathcal{O}(kC)$ [15], where $k$ is the number of items and $C$ is the knapsack capacity. For this work, we reuse the single-threaded, CPU-based implementation of dynamic programming available in the open-source solver SCIP [10]. As we rely on existing implementations, we refer the interested reader to the above references for more details on this algorithm.

Table 1: Statistics for a branch-and-cut run with separation of local cuts for 45 generalized assignment problem instances (left) and 21 instances that were solved to optimization by all variants (right).

| variant | # solved | time | sep. time | #cuts |
|---|---|---|---|---|
| default | 29 | 98.2 | – | – |
| lc0-nc-downlift | 21 | 411.7 | 54.0 | 10858.6 |
| lc0-nc-lifting | 26 | 113.7 | 9.8 | 4933.6 |
| lc1-nc-lifting | 29 | 86.8 | 31.8 | 160244.4 |

| variant | # solved | # nodes | time |
|---|---|---|---|
| default | 21 | 614 | 6.0 |
| lc0-nc-downlift | 21 | 1522 | 33.7 |
| lc0-nc-lifting | 21 | 657 | 5.3 |
| lc1-nc-lifting | 21 | 185 | 4.2 |

## 4.2 The Lifting Routine

Lifting knapsack constraints has been extensively studied in literature, see [43] for a short survey with comprehensive references. Therefore, we only briefly summarize the implemented methods here and refer the interested reader to [43] and references therein for more details.

Let $\{x \in \{0,1\}^n : \sum_{j=1}^n a_j x_j \leq a_0\}$ be one of the original knapsack constraints (corresponding to a single row in $Ax \leq b$ in (MKP)). Define $[n] := \{1, \ldots, n\}$, $F_0 := \{j \in [n] : x_j^\star = 0\}$, and $F_1 := \{j \in [n] : x_j^\star = 1\}$. Then $S := [n] \setminus (F_0 \cup F_1)$ are the variable indices in the reduced knapsack. The lifting procedure then lifts a given inequality $\sum_{j \in S} \alpha_j x_j \leq \alpha_0$ valid for the reduced knapsack polytope $\{x \in \{0,1\}^S : \sum_{j \in S} a_j x_j \leq a_0 - \sum_{j \in F_1} a_j\}$ to a valid inequality for the original knapsack by computing new coefficients $\beta_j$, $j \in F_0 \cup F_1$:

$$\sum_{j \in S} \alpha_j x_j + \sum_{j \in F_1} \beta_j x_j + \sum_{j \in F_0} \beta_j x_j \leq \alpha_0 + \sum_{j \in F_1} \beta_j. \tag{2}$$

We implemented algorithms known as *sequential up-lifting* and *sequential down-lifting*, respectively. The implementation is based on dynamic programming as described by Vasilyev et al. [64].

## 5 Computational Experiments

We implemented the described methods in C/C++, using a developer version of SCIP 8.0.4 (githash 3dbcb38) and CPLEX 12.10 as LP-solver. All tests were performed on a Linux cluster with 3.5 GHz Intel Xeon E5-1620 Quad-Core CPUs, having 32 GB main memory and 10 MB cache. All computations were run single-threaded and with a time limit of one hour. To concentrate on the improvement of local cuts on the dual bound, we initialize all runs with the optimal value. We used $\epsilon = 1 \times 10^{-9}$ in Algorithm 1.

To demonstrate the advantage of using local cuts with the Frank-Wolfe approach, we run our implementation on the generalized assignment instances from the OR-Library available at http://people.brunel.ac.uk/~mastjjb/jeb/orlib/gapinfo.html. These instances have also been used by Avella et al. [4].

The results are presented in Table 1. Here, "default" are the default, factory settings of SCIP. The other settings are lcX-nc-Y, where $X = 0$ means that we only separate local cuts in the root node and $X = 1$ means that we separate local cuts in the whole tree; $Y$ refers to whether we perform down lifting ($Y =$ downlift) or up- and down lifting ($Y =$ lifting). Note that for these settings, where local cuts are enabled, we turn the generation of all other cuts off, because this (somewhat surprisingly) showed better performance. The CPU time in seconds ("time") and separation time ("sep. time") as well as number of nodes ("#nodes") are given as shifted geometric means[3]. The numbers of generated cuts ("#cuts") are arithmetic means. Note that the iteration limit for the Frank-Wolfe algorithm is 10 000 in the root node and 1000 in the subtree. We also reduce the effect of cut filtering allowing for more cuts to enter the main LP. Moreover, we initialize the runs with the best know solution values as in [4].

The results show that the best version is lc1-nc-lifting, i.e., it helps to separate local cuts in every node and perform up- and down lifting. This version is roughly 31% faster than the default settings on the instances solved to optimality. Using only down lifting performs badly. In any case, on these

---

[3]The shifted geometric mean of values $t_1, \ldots, t_r$ is defined as $\left(\prod(t_i + s)\right)^{1/r} - s$ with shift $s = 1$ for time and $s = 100$ for nodes in order to decrease the influence of easy instances in the mean values.

instances, using our local cuts method is a big advantage. Additional results are given in the appendix. In addition, we also ran an experiment in which we applied complemented mixed-integer rounding (CMIR) on the produced cut, which turned out to not be helpful and is therefore not reported in detail.

Some additional observations over all 45 instances in the test set are as follows: Variant `lc1-nc-lifting` called local cuts separation $16\,870.6$ times on average. The total time for Frank-Wolfe separation is about one third of the total time. The time spent in the oracle is 17.3 seconds on average compared to a total of 87.9 seconds for the complete Frank-Wolfe algorithm. On average $69\,968.8$ calls ended running into the iteration limit, 81.2 detected optimality with a zero gradient, 8652.9 stopped because the primal gap is small enough, and $143\,155.9$ stopped because of the termination criterion of Section 3.1.1. This demonstrates the effect of this criterion.

## 6 Conclusion and Future Work

In this paper, we presented a novel method to learn local cuts without relying on solutions of LPs in the process. To show the effectiveness of our approach, we selected the multidimensional knapsack problem as a case study and presented computational results to support our claims.

Solving LPs has proved to be notoriously hard to parallelize, with only minor performance improvements reported in literature to date [40, 45]. Thus, existing methods for deriving local cuts, which rely on solving LPs, typically run single-threaded, on CPUs. Our approach is quite fast, as demonstrated in the computational experiments for our target problem class, but also paves the way for exploring highly parallel implementations on heterogeneous hardware and compute accelerators. This is made possible by eliminating the dependence on LPs and instead relying on the Frank-Wolfe algorithm. One such option we would like to explore in the future is to derive a *vectorized* version of our Frank-Wolfe algorithm that could work on multiple separation problems at the same time, increasing the computational density of the operations performed and availing of massively parallel compute accelerators like *GPUs* in the process.

The presented method is generic and can be applied to any (M)IP. We have chosen one important problem class in this paper to demonstrate the method. A natural extension of this work would be to consider other important problem classes and evaluate the benefits of using our method on those problems - especially those with beneficial properties as outlined in Section 3.2.

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

## Appendix A

We ran two additional experiments. In the first, we try to repeat the experiments about the strength of local cuts in the root node from [47]. As a comparison, we use the implementation of the local cuts with the LP-based approach of [57] (see the beginning of Section 3 for an overview of the method). In the second experiment, we demonstrate what happens if we use local cuts in a branch-and-cut framework to solve multi-dimensional knapsack problems to optimality.

We start with a comparison of the gap-closed, which is defined as $100 - 100\frac{p-d_r}{p-d_{lp}}$, where $p$ is the optimal primal value, $d_r$ is the dual bound at the end of the root node, and $d_{lp}$ is the dual bound of the first LP at the root node. Table 2 shows the results. We use the same multi-dimensional knapsack instances as Kaparis and Letchford: they were originally generated randomly by Chu and Beasley [24] and are available at `http://people.brunel.ac.uk/~mastjjb/jeb/orlib/` `mknapinfo.html`. The instances are organized in blocks of 10, using $n$ variables, $m$ knapsack constraints, and a parameter $\alpha$. Column "KL" shows the gap-closed from [47]. Then the results of using the implementation of the LP-based approach (which is also used in [47]) and our Frank-Wolfe approach are presented. For each approach, we show the average of gap-closed, total running time in seconds, as well as separation time, number of calls, and generated violated cuts by the local cut separation over each instance block (of 10 instances). For both approaches, we turn off all other cuts and strong branching. Moreover, we use settings that allow 1000 rounds of local cuts, 10 000 iterations of the Frank-Wolfe algorithm in the root node, and reduce the effect of cut filtering, i.e., more cuts are added to the LP. In each round we separate local cuts for all knapsack constraints that are available. We also initialize the runs with the optimal value to remove the effects of primal heuristics.

The results show that the LP-based approach achieves similar gap-closed values as Kaparis and Letchford. One explanation for the differences is that the final gap depends on the particular points to be separated (but note that we do not generate rank-2 cuts). In comparison, the Frank-Wolfe approach is much faster, but also produces a smaller gap-closed. There are again several reasons for the differences: We limit the number of Frank-Wolfe iterations to 10 000 in the root node, which will leave some separation problems to be undecided and the Frank-Wolfe algorithm might fail to converge (for instance, for $n = 500$, $m = 5$, $\alpha = 75$, on average 34.8 Frank-Wolfe runs terminated in the iteration limit and 75.4 with the termination criterion explained in Section 2). Moreover, the Frank-Wolfe approach does not necessarily produce a facet, which can weaken the bounds.

Table 3 shows results for running a complete branch-and-cut with variants `lc0-nc-lifting` and `lc1-nc-lifting`. As a comparison, we use the default settings of SCIP, but reduce cut filtering (this produces slightly better results for these instances). The results show that for smaller instances,

Table 2: Gap closed and running times *after the root node* for the multi-dimensional knapsack instances; each line represents the average over 10 instances.

| | | | | LP | | | | | FW | | | | |
|---|---|---|---|---|---|---|---|---|---|---|---|---|---|
| $m$ | $n$ | $\alpha$ | KL | gap closed | time | sepa time | #calls | #cuts | gap closed | time | sepa time | #calls | #cuts |
| 100 | 5 | 25 | 17.96 | 18.88 | 1.60 | 1.60 | 26.1 | 113.0 | 12.54 | 0.77 | 0.76 | 21.3 | 78.4 |
| 100 | 5 | 50 | 21.65 | 22.36 | 2.13 | 2.12 | 29.3 | 124.5 | 15.55 | 0.88 | 0.87 | 20.8 | 77.7 |
| 100 | 5 | 75 | 22.88 | 23.19 | 2.18 | 2.17 | 28.8 | 122.9 | 16.91 | 0.96 | 0.95 | 21.3 | 85.8 |
| 100 | 10 | 25 | 5.55 | 5.75 | 2.69 | 2.68 | 14.5 | 105.7 | 2.73 | 0.80 | 0.79 | 8.0 | 43.0 |
| 100 | 10 | 50 | 7.33 | 7.63 | 3.62 | 3.62 | 16.5 | 120.1 | 4.30 | 1.50 | 1.49 | 11.4 | 59.2 |
| 100 | 10 | 75 | 7.23 | 7.44 | 2.71 | 2.71 | 14.0 | 100.6 | 4.37 | 1.29 | 1.29 | 10.7 | 51.5 |
| 100 | 30 | 25 | 0.16 | 0.14 | 2.89 | 2.87 | 2.4 | 12.0 | 0.00 | 1.05 | 1.05 | 0.2 | 0.1 |
| 100 | 30 | 50 | 0.49 | 0.47 | 4.45 | 4.44 | 4.9 | 38.2 | 0.02 | 0.84 | 0.83 | 0.6 | 0.7 |
| 100 | 30 | 75 | 0.52 | 0.53 | 4.30 | 4.29 | 4.8 | 35.9 | 0.07 | 1.02 | 1.01 | 0.8 | 0.7 |
| 250 | 5 | 25 | 14.56 | 15.44 | 4.03 | 4.00 | 36.3 | 161.0 | 8.03 | 1.08 | 1.06 | 20.4 | 77.6 |
| 250 | 5 | 50 | 15.68 | 14.83 | 6.57 | 6.54 | 38.7 | 172.1 | 7.87 | 1.74 | 1.73 | 20.3 | 78.0 |
| 250 | 5 | 75 | 17.48 | 16.98 | 7.42 | 7.39 | 36.6 | 165.3 | 10.45 | 2.58 | 2.56 | 22.5 | 87.5 |
| 250 | 10 | 25 | 4.53 | 3.98 | 6.17 | 6.15 | 17.5 | 128.7 | 1.83 | 1.51 | 1.50 | 10.4 | 49.7 |
| 250 | 10 | 50 | 4.48 | 4.11 | 6.52 | 6.48 | 17.4 | 130.8 | 1.60 | 1.49 | 1.48 | 8.4 | 46.2 |
| 250 | 10 | 75 | 5.03 | 4.73 | 8.54 | 8.51 | 18.5 | 141.3 | 2.10 | 2.12 | 2.11 | 9.5 | 51.4 |
| 500 | 5 | 25 | 13.80 | 11.75 | 10.37 | 10.31 | 42.0 | 182.9 | 5.96 | 2.84 | 2.81 | 20.9 | 78.4 |
| 500 | 5 | 50 | 11.91 | 11.17 | 17.41 | 17.35 | 45.2 | 199.2 | 5.02 | 5.21 | 5.19 | 21.5 | 84.0 |
| 500 | 5 | 75 | 13.70 | 11.75 | 22.53 | 22.48 | 41.5 | 190.0 | 5.62 | 7.63 | 7.60 | 22.4 | 85.4 |

Table 3: Detailed statistics for a branch-and-cut run with three different algorithm variants for the multi-dimensional knapsack instances; each line represents shifted geometric means over 10 instances.

| | | | default | | | lc0-nc-lifting | | | lc1-nc-uplift | | |
|---|---|---|---|---|---|---|---|---|---|---|---|
| $n$ | $m$ | $\alpha$ | #solved | time | sep time | #solved | time | sep time | #solved | time | sep time |
| 100 | 5 | 25 | 10 | 11.76 | 5.11 | 10 | 5.89 | 1.45 | 10 | 24.41 | 18.95 |
| 100 | 5 | 50 | 10 | 7.61 | 3.32 | 10 | 5.27 | 1.57 | 10 | 19.62 | 16.00 |
| 100 | 5 | 75 | 10 | 4.88 | 2.39 | 10 | 3.67 | 1.70 | 10 | 9.62 | 7.74 |
| 100 | 10 | 25 | 10 | 47.49 | 10.83 | 10 | 32.67 | 2.27 | 10 | 347.27 | 308.99 |
| 100 | 10 | 50 | 10 | 44.48 | 11.49 | 10 | 30.04 | 2.77 | 10 | 360.69 | 325.67 |
| 100 | 10 | 75 | 10 | 16.44 | 5.20 | 10 | 11.33 | 2.80 | 10 | 112.23 | 102.36 |
| 100 | 30 | 25 | 10 | 558.98 | 18.90 | 10 | 458.22 | 2.07 | 3 | 3288.89 | 3095.03 |
| 100 | 30 | 50 | 10 | 497.33 | 20.35 | 10 | 413.70 | 1.63 | 1 | 3057.23 | 2870.52 |
| 100 | 30 | 75 | 10 | 118.77 | 12.38 | 10 | 92.87 | 1.91 | 7 | 1986.30 | 1892.20 |
| 250 | 5 | 25 | 10 | 83.72 | 12.55 | 10 | 63.59 | 2.50 | 10 | 267.26 | 195.77 |
| 250 | 5 | 50 | 10 | 125.84 | 10.07 | 10 | 104.14 | 3.54 | 10 | 636.25 | 493.87 |
| 250 | 5 | 75 | 10 | 53.20 | 7.89 | 10 | 42.53 | 4.08 | 10 | 231.73 | 188.66 |
| 250 | 10 | 25 | 0 | 3600.01 | 906.72 | 2 | 3419.56 | 4.13 | 0 | 3600.01 | 2992.23 |
| 250 | 10 | 50 | 1 | 3528.15 | 1033.57 | 1 | 3483.90 | 4.03 | 0 | 3600.00 | 3326.01 |
| 250 | 10 | 75 | 6 | 2062.86 | 346.76 | 7 | 1645.50 | 5.10 | 1 | 3482.48 | 3138.87 |
| 500 | 5 | 25 | 8 | 1883.73 | 218.06 | 10 | 1539.94 | 5.07 | 3 | 3056.45 | 1962.73 |
| 500 | 5 | 50 | 7 | 1422.77 | 206.38 | 10 | 1083.33 | 7.73 | 5 | 2963.75 | 2068.72 |
| 500 | 5 | 75 | 9 | 676.14 | 70.40 | 10 | 548.03 | 9.43 | 7 | 2039.17 | 1368.61 |

there is no clear advantage of the Frank-Wolfe approach, but for larger sizes, it solves more instances and is faster.

