# OpenReview forum: "Learning Cuts via Enumeration Oracles"
_NeurIPS.cc/2023/Conference — NeurIPS 2023 poster_

### Official Review · Reviewer_avng · 2023-06-23

**Soundness:** 4 excellent
**Presentation:** 4 excellent
**Contribution:** 2 fair
**Rating:** 6
**Confidence:** 3

**Summary:**

The paper proposes an efficient algorithm for learning local cuts used in integer programming relaxations. The algorithm uses a variant of the Frank-Wolfe algorithm and employs a stopping criterion for reducing the number of iterations during the application of the Frank-Wolfe algorithm. The paper showcases the effectiveness of the proposed algorithm by applying it to the multidimensional knapsack problem and reports its findings.

I have read the authors' rebuttal.

**Strengths:**

* The paper contributes to solving the generic IP problem efficiently.
* The paper includes experimental results showing the usefulness of the proposed algorithm.

**Weaknesses:**

The paper lack a discussion about the relevant of the studied problem to the NeurIPS community. I am not sure how much the results would appeal to the broad NeurIPS audience.

**Questions:**

Please include a discussion about the relevance of the problem to NeurIPS audience. I appreciate the theoretical work, but I am not sure about its relevance to the broad  NeurIPS audience. At least this warrants further discussion and motivation, which are currently lacking from the paper.

**Limitations:**

Yes

---

> ### Author Rebuttal · Authors · 2023-08-09
>
>
> 1. The paper lack a discussion about the relevant of the studied problem to the NeurIPS community. I am not sure how much the results would appeal to the broad NeurIPS audience. Please include a discussion about the relevance of the problem to NeurIPS audience. I appreciate the theoretical work, but I am not sure about its relevance to the broad NeurIPS audience. At least this warrants further discussion and motivation, which are currently lacking from the paper.
>
> We understand the concern about the direct relevance of our work to the broader NeurIPS audience. Although the intersection of learning methods and integer problem-solving might not be a mainstream application in Machine Learning, there has been a rising trend of studies in this area, spurred by advancements in learning techniques. In fact, several past NeurIPS publications align with our topic. Some of these include:
>
> - Wu, Y., Song, W., Cao, Z., Zhang, J., Gupta, A., & Lin, M. S. (2022). Graph Learning Assisted Multi-Objective Integer Programming. In NeurIPS.
> - Chmiela, A., Khalil, E., Gleixner, A., Lodi, A., & Pokutta, S. (2021). Learning to Schedule Heuristics in Branch-and-Bound. In NeurIPS.
> - Wu, Y., Song, W., Cao, Z., & Zhang, J. (2021). Learning large neighborhood search policy for integer programming. In NeurIPS.
> - Chen, X., & Tian, Y. (2019). Learning to perform local rewriting for combinatorial optimization. In NeurIPS.
> - He, H., Daume III, H., & Eisner, J. M. (2014). Learning to search in branch and bound algorithms. In NeurIPS.
>
> This point is also reinforced by the review of reviewer kHh3 (see "Strengths").
>
> Given this context, we are confident that our interdisciplinary exploration holds value for the community. Such work could stimulate knowledge exchange and potentially motivate machine learning enthusiasts to devise novel applications or enhancements in optimization problems using machine learning techniques. Acknowledging your feedback, we agree that the paper should offer more clarity on the significance of employing learning methods in integer programming. To this end, we will add a dedicated paragraph in our paper's revised version to elucidate this relationship further.

---

> > ### Comment · Reviewer_avng · 2023-08-16
> > **Response acknowledgment**
> >
> > I thank the authors for their responses. My rating remains the same.

---

### Official Review · Reviewer_XrPs · 2023-07-03

**Soundness:** 3 good
**Presentation:** 4 excellent
**Contribution:** 4 excellent
**Rating:** 7
**Confidence:** 4

**Summary:**

The authors propose a new method for generated lifted cuts from a smaller projected polytope that replaces the usually-expensive LP/IP-based separation routines with an optimization-based method that uses the Frank-Wolfe algorithm. The authors present theory showing that the proposed separation routine indeed produces strong cuts. They then validate their methods via a case study on the multidimensional knapsack problems and variants.

**Strengths:**

The proposed method is novel and is a genuinely new and interesting algorithmic development in cutting plane generation. The paper is written clearly and precisely. The need for faster separation routines that do not rely on large LPs (or even MIPs) requiring expensive column generation routines is clear, and the authors make interesting progress towards a solution.

**Weaknesses:**

I think the Knapsack case study could’ve used some more details, including comparisons to existing methods of lifting in this domain. E.g., how does the authors’ Frank-Wolfe based method compare to the usual pipelines of lifting for knapsack inequalities – in particular sequential up-lifting/down-lifting for minimal cover inequalities? Also, how does the authors’ separation procedure itself compare to the usual separation routines for cover inequalities (e.g., those in “Lifted Cover Inequalities for 0-1 Integer Programs: Computation” by Gu, Nemhauser, and Savelsbergh)? Does it produce similar cuts? Stronger ones? Weaker ones? Is it faster or slower? Knapsack constraints are one of the main areas where lifting really shines since it can be done fairly efficiently, so some more comparison here would be super interesting. But, I still view this paper as providing a nice contribution without the deeper dive into lifting knapsack cover inequalities.

A comparison to the equivalent local cuts implementation that SCIP uses seems important to include, but maybe SCIP doesn’t do anything like this? Tangentially, to what extent are local cuts actually used by general-purpose MIP solvers?


**Questions:**

Methodologically, how does the (high-level) approach differ from lift-and-project? Some brief discussion of the similarities/differences would be nice to include.

Do the authors think this approach would scale beyond the kinds of multi-dimensional knapsack/GAP problems studied? Those problems conveniently have efficient lifting methods, but for general MIP lifting can be expensive (requiring solutions to MIPs). (I do understand that lifting is not the focus of this paper.)

---

> ### Author Rebuttal · Authors · 2023-08-09
>
> Comments on weakness:
> 1. [...] E.g., how does the authors’ Frank-Wolfe based method compare to the usual pipelines of lifting for knapsack inequalities – in particular sequential up-lifting/down-lifting for minimal cover inequalities?  [...] Also, how does the authors’ separation procedure itself compare to the usual separation routines for cover inequalities (e.g., those in “Lifted Cover Inequalities for 0-1 Integer Programs: Computation” by Gu, Nemhauser, and Savelsbergh)? Does it produce similar cuts? Stronger ones? Weaker ones? Is it faster or slower? [...]
>
> The mentioned knapsack separation routines are applied in the default settings of SCIP. Since SCIP is considered state-of-the-art for the open source solvers, we assume that its internal cut selection mechanism would prefer these cuts if they were helpful.
> While SCIP is very informative about how many separators were called and how many cuts were effectively applied, we did not analyze this further in the paper, as a discussion here would quickly evolve out of the scope of the paper. However, we would gladly publish all the SCIP logs from the experiments alongside the revised manuscript.
>
>
> 2. A comparison to the equivalent local cuts implementation that SCIP uses seems important to include, but maybe SCIP doesn’t do anything like this? Tangentially, to what extent are local cuts actually used by general-purpose MIP solvers?
>
> SCIP currently dooes not include an implementation of either local cuts method (see, e.g. https://www.scipopt.org/doc/html/group__SEPARATORS.php for a list of cuts available). Hence, our experiments focus on comparing our FW-local cuts to "all that SCIP has to offer", i.e., the default configuration which permits all cuts subject to SCIP's built-in cut selection mechanism.
>
> Answers to questions
>
> 1. Methodologically, how does the (high-level) approach differ from lift-and-project? Some brief discussion of the similarities/differences would be nice to include.
>
> We appreciate your suggestion and agree that a comparative discussion would be useful. Lift-and-project methods share some high-level ideas with our approach, such as exploring solutions in a different space and then projecting back to the original space. However, their mechanics are significantly different. Lift-and-project methods convert the problem, e.g., into an equivalent quadratic formulation to simulate non-linearity in integer programs, which is then linearized in the lifting phase. Afterward, this extended formulation is projected onto the original variable space, resulting in a strengthened binary MIP formulation. This step is the projection phase. In contrast, our approach, maintains the linearity of the program. We work with a trivial projection of the polytope onto a lower-dimensional variable space where the problem can be solved more efficiently, and then the solution is lifted back to the full variable space. We will elaborate on this comparison in the revised manuscript.
>
> 2. Do the authors think this approach would scale beyond the kinds of multi-dimensional knapsack/GAP problems studied? Those problems conveniently have efficient lifting methods, but for general MIP lifting can be expensive (requiring solutions to MIPs). (I do understand that lifting is not the focus of this paper.)
>
> We recognize that the effectiveness of our approach partly relies on the availability of efficient lifting routines, which might not be the case for all problem classes. However, the final impact of our method on the overall solving process involves a tradeoff between the runtime of the lifting routine and the strength of the produced cuts, both of which can vary with the problem class. Thus, even problem classes with less efficient lifting routines might theoretically benefit from our method. Many significant and widely applicable problem classes do possess efficient lifting methods and local cuts have been successfully applied to them (as referenced in the "Related Work" section). For some problems, no lifting is required at all (for instance, when the Trivial Lifting Lemma holds, see Section 3.2), such as with the Linear Ordering Problem. We did not conduct thorough experiments on other problem classes, but this is a part of our planned future work. It is challenging to make confident predictions due to the numerous factors involved in evaluating MIP/IP solver performance.

---

> > ### Comment · Reviewer_XrPs · 2023-08-16
> >
> > I actually couldn't find anything about cover cuts/lifting in the SCIP separators link (but maybe I didn't look hard enough). Nonetheless, I thank the authors for their response and maintain my rating for the paper to be accepted.

---

> > > ### Author Response · Authors · 2023-08-21
> > >
> > > Dear Reviewer,
> > >
> > > here's a discussion in which one of the SCIP developers confirms that cover cuts are separated automatically:
> > >
> > > http://listserv.zib.de/pipermail/scip/2020-April/003925.html

---

### Official Review · Reviewer_kHh3 · 2023-07-05

**Soundness:** 3 good
**Presentation:** 4 excellent
**Contribution:** 3 good
**Rating:** 7
**Confidence:** 4

**Summary:**

This paper presents a method for generating separating cutting planes for integer programming problems that is based on the Frank-Wolfe method for optimizing over polyhedron. It falls roughly within the existing local/Fenchel cut framework: given a point, it identifies the closest point within the feasible region by repeatedly calling a (hopefully fast) optimization oracle, and then

**Strengths:**

Overall I quite like the paper and how it stitches together some disparate, existing ideas from the optimization literature.

The paper is generally very well-written. The paper is in an area of clear interest to the Operations Research and Mathematical Optimization community, and the subject matter is consistent with a number of other papers accepted to NeurIPS in the past. The application of Frank-Wolfe in this manner is interesting and non-trivial, as the authors apply some extensions to improve its practical performance (the "FW Gap" characterization being chief among them).

**Weaknesses:**

The results in the computational study are slightly underwhelming. A 31% improvement in solving times is nice, but this only considers "solved" instances and the geo-mean solve times here are quite small to begin with (6s for "default"). The provided solve times on all instances (including ones that terminate without proving optimality) show a significantly smaller (though still positive) speedup. Given that MIP solvers such as SCIP tend to be engineered more for hard instances than easy ones, it is difficult to know how much to read into these results without additional detail.

**Questions:**

* L19: "However, the by far most interesting case is the one we consider here" (IP vs. MIP). This is a subjective statement that I imagine many researchers in the MIP community would (strongly!) disagree with. I'd suggest softening or removing, as this is not really necessary to justify the restriction to IP for the purposes of the paper.
* L30: "first p indices" seems like it is left over from an earlier draft that considered MIP.
* The first paragraph of Section 2 is fairly confusing, largely because you introduce two polyhedron (P and tilde{P}), but really only work with the second. As such, statements about "valid cut"s and "original space" are unclear without further qualification. I'd suggest rewriting it, and maybe adding some discussion about how P and tilde{P} should or do relate to each other: "subproblem" suggests some relationship (maybe containment in a projected space?), but is vague.
* L101: You do not assume that tilde{P} is full-dimensional (just P, but see comment above), so there may not exist an interior point.
* Figure 1: The figure and caption are a bit confusing: it's not clear what the tilde{P}_I label refers to; indices t and k are used variously in the caption and figure; there is no explicit reference to (c) in the caption, etc.
* L151: Is the oracle required to return a solution vertex? If so, you should state that above. If not, does this affect any of the statements to follow?
* L317: It is a bit confusing to the reader that you state a percentage solve time improvement in an unqualified manner, when it only applies to a subset of the instances in your test bed (the solved ones). Please be more explicit about this in the test (here, and also earlier in the contributions section).
* Section: To bolster the takeaways from the computational study, it would be interesting to have more information about the relative performance on the unsolved instances. For instance: How many instances does each method solve to optimality within the time budget? What do the dual and primal/dual integrals look like for the various methods? How long does "default" SCIP spend in cut separation? A separate suggestion to isolate the dual effect of the new method would be to seed each method with the best known primal solution as a warm start.

**Limitations:**

Yes.

---

> ### Author Rebuttal · Authors · 2023-08-09
>
>
> 1. L19: "However, the by far most interesting case is the one we consider here" (IP vs. MIP). This is a subjective statement that I imagine many researchers in the MIP community would (strongly!) disagree with. I'd suggest softening or removing, as this is not really necessary to justify the restriction to IP for the purposes of the paper.
>
> We understand the concern and agree that the phrasing needs refinement. Our objective wasn't to compare IP vs. MIP directly, but rather to highlight that the methods proposed in our paper are more readily applicable to the IP case. We will amend this in the revised manuscript.
>
> 2. L30: "first p indices" seems like it is left over from an earlier draft that considered MIP.
>
> Indeed, that is an oversight on our part, a remnant from considering the MIP case. We will revise this in the updated manuscript. Thanks!
>
> 3. The first paragraph of Section 2 is fairly confusing, largely because you introduce two polyhedron (P and tilde{P}), but really only work with the second. As such, statements about "valid cut"s and "original space" are unclear without further qualification. I'd suggest rewriting it, and maybe adding some discussion about how P and tilde{P} should or do relate to each other: "subproblem" suggests some relationship (maybe containment in a projected space?), but is vague.
>
> We appreciate your feedback and will work on providing a clearer relationship between the two polyhedra in the revised manuscript. tilde{P} can be thought of as a projection of P onto a lower-dimensional space.
>
> 4. L101: You do not assume that tilde{P} is full-dimensional (just P, but see comment above), so there may not exist an interior point.
>
> Thank you for pointing this out. We indeed have to assume that the projection is full-dimensional, which we will do in the revised manuscript.
>
> 5. Figure 1: The figure and caption are a bit confusing: it's not clear what the tilde{P}_I label refers to; indices t and k are used variously in the caption and figure; there is no explicit reference to (c) in the caption, etc.
>
> Thanks for pointing this out! We will definitely fix the figure in the revised version of the manuscript. While the figure itself is correct,
> the caption does indeed contain several errors; for explanation:
> - (c) shows the iteration "at convergence", the reference should be in the last sentence of the caption;
> - the index "t" should actually be "i" from Figure (c). Due to our assumptions, we know that FW converges, so for every iteration k before convergence, there exists an index i > 0 such that convergence/termination occurs at iteration k + I.
>
> Lastly, as we are allowed to provide one PDF page worth of figures and tables as part of our "global" rebuttal, we attach the updated figure there for your reference.
>
> 6. L151: Is the oracle required to return a solution vertex? If so, you should state that above. If not, does this affect any of the statements to follow?
>
> Thanks for raising that point. In our setting and example (MKP), the oracle indeed always returns a vertex, but this is not a requirement for the following statements, especially for the theory of FW (the property is never explicitly used). We will clarify this in the revised version.
>
> 7. It is a bit confusing to the reader that you state a percentage solve time improvement in an unqualified manner, when it only applies to a subset of the instances in your test bed (the solved ones). Please be more explicit about this in the test (here, and also earlier in the contributions section).
>
> Your point is well-taken, and we apologize for any confusion. We will revise the manuscript to explicitly state what the percentage refers to.
>
> 8. Section: To bolster the takeaways from the computational study, it would be interesting to have more information about the relative performance on the unsolved instances. For instance: How many instances does each method solve to optimality within the time budget? What do the dual and primal/dual integrals look like for the various methods? How long does "default" SCIP spend in cut separation? A separate suggestion to isolate the dual effect of the new method would be to seed each method with the best known primal solution as a warm start.
>
> We appreciate your feedback. For all full branch-and-bound run results, the number of instances each method solved within the time limit is listed (see Tables 1 and 3). Additionally, we already exclude the effect of primal heuristics on the solution process by initializing the runs with the optimal solution values.
> This fact is only mentioned in the Appendix - we will move this point to the main part of the paper and include details on time spent in cut separation in the revised version of the paper.

---

> > ### Comment · Area_Chair_Yeki · 2023-08-20
> >
> > Dear Reviewer,
> >
> > The author-reviewer discussion period is closing soon, so could you please go over the authors' rebuttal and respond with a message to the authors? It is important that authors receive a reply to their rebuttals, as they have tried to address comments raised by the reviewers.
> >
> > Best regards,
> > AC

---

### Official Review · Reviewer_5Xhs · 2023-07-14

**Soundness:** 3 good
**Presentation:** 4 excellent
**Contribution:** 3 good
**Rating:** 6
**Confidence:** 4

**Summary:**

* Context:
This paper deals with the subject of generating cuts for solving Integer Programs (MIP).
Rather than using a cut generating algorithm based on a formula (like Gomory cuts) to create hyperplanes separating the solution of the relaxed problem from the (integer) feasible domain, the papers attempts to generate "Local cuts".
Local cuts are cuts that are obtained by generating a cut in a subproblem with reduced dimension (so that any optimization problem required to generate the cut are easier to solve), and then, taking the cut for the subproblem and transforming it into a cut for the actual problem to solve.

* Contribution:
This paper presents a local cuts generating algorithm, based on the Frank-Wolfe / conditional gradient algorithm.
Frank-Wolfe is used to find the point in the subproblem polyhedron $\tilde{P}$ that is the closest to the point to be separated $\tilde{x}$, as this point will be likely to be on a facet.
The Frank-Wolfe algorithm has the advantage that it only require linear optimization oracle access over the subproblem to arrive at a solution, which can often be implemented efficiently.
The paper also demonstrates that using duality information, the Frank-Wolfe algorithm does not need to be run all the way to convergence and that the strength of the cut generated by intermediate iterates can be evaluated, leading to a speed-up.

The impact of using these cuts is evaluated on the multi-dimensional knapsack problem, by including the local cut generating algorithm into the open source SCIP solver.

**Strengths:**

The paper presents a algorithmic improvement for solving an important problem, and does it in a clear fashion.
It is clearly delineated what exists in the current literature and what constitutes an actual contribution of the work.
Explanations and motivations are straightforward to follow.

**Weaknesses:**

* From reading the paper, what is not obvious is how to take the contribution of this paper, and apply it to a different problem. After reading the paper, I understand how to apply this to a knapsack problem, but it's not obvious to me how I would do it if I had a more arbitrary MIP (for example one for verifying neural networks), or even a different type of problem solvable by integer programs (let's say a TSP). What would be beneficial would be to have somewhere clearly "These are the requirements that you need for your problem to be solvable using this method"

* I think that what is missing is some appropriate baseline from the evaluation. As it is, the contribution of the paper seems to me to be "here is an efficient way to do local cuts", while the only baseline that is compared to is default SCIP. Is the performance difference that is seen due to using local cuts or using the local cuts presented in this paper? (I think that Table 3 in the appendix might have some results that are relevant to this, but this would benefit from being moved to the main part of the paper.)

**Questions:**

* [Section 2] Could you clarify the relation between the polytope $P \subseteq \mathbb{R}^n$ and the polyhedron $\tilde{P} \subseteq \mathbb{R}^k$ ? Is it always a restriction (a subset of the variables in P, such that an element is in $\tilde{P}$ if there exists an element in $P$ with those values?) or could it be something more general ( like an affine transformation of $\tilde{P}$ ?) and have as only requirement "if we have a cut in $\tilde{P}$, we can find a cut in $P$? . As it is, it sounds quite abstract.

* [Section 5] Downlifting vs up- and down lifting is not explained anywhere. Given that downlifting performs significantly worse on all benchmarks, it seems that it might not be worth it to include it in the main version of the paper to avoid confusion.
Similarly, CMIR seems to have very little impact so unless it can be explained, it might make sense to move it to the appendix.
* Would it be possible to include some more information on what the default settings of SCIP looks like? Is it a reasonable baseline? Is there an indication on how many cuts (whether local or not local) it corresponds to?

Typos / small comments:
- Line 30, I don't understand what the notion of the first $p$ indices is. Is that a reference to some Mixed Integer Programming case, where only the first $p$ variables need to be integral?
- Figure 1 legend, is it supposed to be the L_2 projection (rather than L_t) ?

**Limitations:**

Yes.

---

> ### Author Rebuttal · Authors · 2023-08-09
>
> Comments on weaknesses:
>
> 1. [..] What would be beneficial would be to have somewhere clearly "These are the requirements that you need for your problem to be solvable using this method" [..]
>
> As we restrict the scope to IPs for this paper, we will only list the requirements for applying the method for IPs. To be able to apply the method in general, you'll need the following components:
> a) A projection P -> \tilde{P}. Note that we restrict \tilde{P} to always be a subset of P, which restricts the degrees of freedom. In order to be practical, the selected \tilde{P} should be such that the oracle (see 2.) solves linear optimization problems over \tilde{P} efficiently.
> b) An oracle solving problems over \tilde{P} to optimality (returning a vertex). Note that while there may exist practically efficient specialized methods (such as dynamic programming for MKP), enumeration constitutes a general go-to option, especially in the presence of parallel hardware.
> c) A lifting method to lift cuts from \tilde{P} up to P.
>
> Nevertheless, we think making this list explicit is a great idea. While all components are there, we will make sure that in the revised version, we modify the language accordingly to clarify these conditions.
>
> 2. [...] while the only baseline that is compared to is default SCIP. Is the performance difference that is seen due to using local cuts or using the local cuts presented in this paper? [..]
>
> In Table 1, we compare the performance of baseline SCIP (aka "default") to variants of SCIP where the only cuts enabled are our FW-Local-Cuts (lc0 if separation only occurs in the root node; lc1 if this is done in the whole tree). As the reviewer correctly notes, the desired computation is indeed shown in the appendix, but here in Table 2: There, we compare _our_ local cuts (FW) with the LP-based local cuts from [42].
>
> Note that both methods result in different cuts, hence one always has to take both the runtime as well as the gap closed measure into account when comparing them. The third paragraph in Appendix 2 discusses the results.
>
> Answers to questions:
>
> 1. [Section 2] Could you clarify the relation between the polytope P and the polyhedron P_tilde? Is it always a restriction (a subset of the variables in P, such that an element is in if there exists an element in with those values?) or could it be something more general (like an affine transformation of?) and have as only requirement "if we have a cut in, we can find a cut in? As it is, it sounds quite abstract.
>
> In the paper, we consistently assume that \(P_{tilde}\) is a subset of P, with no transformations of any kind. This will be clarified in the revised version. We have not delved into the potential application of affine transformations. While they might be theoretically possible, they would greatly complicate the process of efficiently lifting cuts to the original space. Moreover, we did not discern a distinct advantage in applying such transformations.
> The restriction to using subsets of variables only has, in our opinion, already enough degrees of freedom (think, e.g., projecting out continuous variables in a MIP) yet enables us to find amenable subproblems quite effectively, as shown, e.g., in the case study for MKP.
>
> 2. [Section 5] Downlifting vs up- and down lifting is not explained anywhere. Given that downlifting performs significantly worse on all benchmarks, it might not be worth it to include it in the main version of the paper to avoid confusion. Similarly, CMIR seems to have very little impact so unless it can be explained, it might make sense to move it to the appendix.
>
> Lifting procedures are inherently problem-specific. The lack of detailed explanations for individual lifting procedures for MKP in the paper stems from our reliance on established methods from existing literature, for which we provide references. We initially included all variants in the primary experiments to align with the customary practices in MKP literature. However, acknowledging your point, we are inclined to relocate the less effective variants to the appendices in the revised version to enhance clarity.
>
> 3. Would it be possible to include some more information on what the default settings of SCIP look like? Is it a reasonable baseline? Is there an indication on how many cuts (whether local or not local) it corresponds to?
>
> In general, we understand the desire to include a thorough discussion here; nevertheless, we also see the degree of complexity that would be required to correctly reflect this in the paper. For example, the default settings in SCIP enable the use of all implemented types of cuts, but the actual application is subject to an internal cut selection procedure. While SCIP is very informative in this regard, e.g. writing out how many separators were called and how many cuts were effectively applied, a discussion here would quickly evolve out of the scope of the paper. SCIP is considered best-in-class for the academic solvers, hence the default settings usually reflect a decent baseline.
> Finally, we could offer a compromise here: Together with the revised version of the paper, we publish the SCIP log files of all the experiments contained in the paper which contain the data.
>
> 4. Line 30, I don't understand what the notion of the first indices is. Is that a reference to some Mixed Integer Programming case, where only the first variables need to be integral?
>
> Your observation is accurate. In the context of MIP, the terminology would be applicable, but our focus on the IP case, where every variable is integral, requires a different explanation. In the IP scenario, LP-relaxations are undertaken as long as the solution \(x^*\) has fractional values across one of the n indices. We will rectify this in the revised paper.
>
> 5. Figure 1 legend, is it supposed to be the L_2 projection (rather than L_t) ?
>
> You're right; it should indeed be L_2. This oversight will be addressed in our paper's next version.

---

> > ### Comment · Reviewer_5Xhs · 2023-08-11
> >
> > Thank you for your detailed response, notably for pointing me to the appendix. Table 2 there is a bit hard to interpret, given the fact that between LP generated cuts and FW generated cuts, there is nothing held constant (which could be either time until a given gap closed, or setting a time budget and reporting the gap closed, although I imagine this might be hard to setup in practice.)
> >
> > I still found the paper interesting enough as it is, and am happy to maintain my rating and for this paper to be accepted.

---

> > > ### Author Response · Authors · 2023-08-15
> > >
> > > Thank you for your comment and maintaining the rating! We will try and figure out a way to improve Table 2 in the revised version. At the very least, we will try and report the separation time for the LP-based approach as well, making it easier to compare against the FW-case.

---

### Official Review · Reviewer_bBeY · 2023-07-20

**Soundness:** 2 fair
**Presentation:** 2 fair
**Contribution:** 2 fair
**Rating:** 3
**Confidence:** 4

**Summary:**

The paper studies cut generation from operations research. While this is potentially a promising approach, I believe the paper falls out of scope of the conference. Optimization can be seen as part of machine learning machiery, but normally NeurIPS papers are expected to have some connection to machine learning, e.g., at the very least, the experiments could have been done on some machine learning problem. It is not common to reject a paper based on out of scope, however in this case the paper is purely operations research. I believe submiting to an operations research venue would be more appropriate, and it is likely that the impact of the work would be greater in such a community.

**Strengths:**

Did not look in detail.

**Weaknesses:**

Did not look in detail.

**Questions:**

What is the motivation for submiting this paper to NeurIPS?

**Limitations:**

Did not look in detail.

---

> ### Author Rebuttal · Authors · 2023-08-09
>
>
> 1. What is the motivation for submiting this paper to NeurIPS?
>
> We recognize that our paper primarily falls under Operations Research, but it's crucial to note a significant distinction. Unlike many conventional cutting plane techniques in this field that hinge on fixed equations and formulas for separating hyperplane derivation, our method aims to learn unknown facets of a polytope from a lower-dimensional version, subsequently generalizing them into cutting planes for the high-dimensional polytope. This learning approach, while not new to Operations Research as high-level meta-paradigm, is made tangible and practical by our novel use of the Frank-Wolfe algorithm. This innovation paves the way for *general-purpose* cutting planes, learned directly from the problem rather than from pre-established rule sets.
>
> We understand that this may not be the most typical application of learning algorithms, yet we respectfully differ with the view that it lacks interest for the community. We believe it could instead stimulate knowledge exchange across fields and potentially inspire machine learning researchers to apply new or improved methods derived from machine learning advancements to optimization problems.
>
> Furthermore, several NeurIPS papers from previous years have explored the application of learning methods in solving integer programming problems. A few notable ones are:
>
> - Wu, Y., Song, W., Cao, Z., Zhang, J., Gupta, A., & Lin, M. S. (2022). Graph Learning Assisted Multi-Objective Integer Programming. In NeurIPS.
> - Chmiela, A., Khalil, E., Gleixner, A., Lodi, A., & Pokutta, S. (2021). Learning to Schedule Heuristics in Branch-and-Bound. In NeurIPS.
> - Wu, Y., Song, W., Cao, Z., & Zhang, J. (2021). Learning large neighborhood search policy for integer programming. In NeurIPS.
> - Chen, X., & Tian, Y. (2019). Learning to perform local rewriting for combinatorial optimization. In NeurIPS.
> - He, H., Daume III, H., & Eisner, J. M. (2014). Learning to search in branch and bound algorithms. In NeurIPS.
>
> This point is also reinforced by the review of reviewer kHh3 (see "Strengths").

---

> > ### Comment · Reviewer_bBeY · 2023-08-15
> >
> > My issue is still that, to me, the paper seems to clearly fall within Operations Research without an obvious connection to machine learning.
> >
> > I agree that combining optimization and (machine) learning is a promising research direction. However the "learning" component in this work seems to be different from what I would normally expect from papers combining machine learning and optimization.
> >
> > To make this more concrete, consider the third paper you referenced, namely "Learning large neighborhood search policy for integer programming" by Wu et al. The paper is about using deep reinforcement learning (machine learning) to solve integer programs. When reinforcement learning faces integer programs, this gives rise to many technical difficulties given the nonlinearity of integer programs, and this is interesting to explore how to learn effective policies given these very difficult challenges. There is a clear connection to machine learning in this case, and many researchers from reinforcement learning and optimization could find interest in the work. Other works dealing with optimization and machine learning might find related difficulties in generating data for learning, selecting appropriate metrics for learning, etc.
> >
> > But in your paper, the "learning" is rather on the OR side, and I do not see this connection to the broader machine learning community. Can you point out specifically which part of the paper could potentially be of interest to machine learning researchers (even in a broad sense)?

---

> > > ### Author Response · Authors · 2023-08-21
> > >
> > > We appreciate the comments and point of view of the reviewer and would like to complement them with ours: Traditionally cutting planes have been derived from polynomial time verifiable methods. This limits their expressive power. We take a different approach by considering subproblems of an NP-hard problem (which theoretically inherits its complexity status), generate cuts and lift them. The process of obtaining a potentially polynomial non-verifiable cutting plane and generalizing it to higher dimension we call learning as the structural insights from lower dimension that are encoded in the cutting plane are applied to 'before unseen' higher dimension versions of the problem. This learning process has a clear polyhedral interpretation and we demonstrate that it is useful to collect enough knowledge about the problem to speed-up the solution process. Moreover, we think that there is a certain overlap between the learning and mathematical optimization community (or Operations Research, if you like). We see our paper in this area.

---

> > > > ### Comment · Reviewer_bBeY · 2023-08-21
> > > >
> > > > Thank you, your response has been noted. I will discuss this with the other reviewers.

---

### Official Review · Reviewer_cm37 · 2023-07-26

**Soundness:** 2 fair
**Presentation:** 3 good
**Contribution:** 2 fair
**Rating:** 5
**Confidence:** 3

**Summary:**

This paper studies integer programming (IP) by proposing an alternative cutting-plane approach that makes use of the Frank-Wolfe (FW) algorithm for the separation sub-routine (i.e., separation of the target from the feasible set). The main idea is to use “local cuts” which aims at deriving the facets of $P_I$, i.e., the integer hull of the feasible set, $P$, of the relaxed linear program (LP).

Facets of $P_I$ are the strongest cuts but they are unknown and finding the facets could be computationally expensive. The authors propose a separation procedure which does not need to solve an LP, and instead uses away-step FW variant to (approximately) solve the separation problem.

To demonstrate the performance of their approach, the authors compare their FW-based approach with the default SCIP solver, an optimization framework for solving mixed-integer programs, against solving the multi-dimensional knapsack problem. They also describe the implementation of the linear minimization oracle (LMO) of the FW algorithm using dynamic programming for the application at hand.

**Strengths:**

-	Authors provide an alternative to the cutting-plane approaches by simplified local cuts. Instead of exactly solving an LP, the authors make use of FW algorithm and dynamic programming to find facets of the underlying polyhedron in lower dimensions. They use known lifting techniques to obtain the full-dimensional cut.


-	The stopping criterion for the algorithm is derived via fundamental properties of the FW algorithm and LMOs. Using a fundamental, technical intuition, they improve the performance of the separation routine by an additional stopping criterion.


-	They compare their approach with SCIP with default settings as a benchmark and provide improved run times.

**Weaknesses:**

-	The fundamental differences and improvements with respect to existing cutting-plane methods to solve IP problem (specifically MKP as the authors provide it as an example) are not clarified. Computational/complexity improvements with respect to LP-based methods are not studied in technical details.


-	The method is not comprehensively compared against (in practice) to existing methods for solving IPs, specifically MKPs. The authors provide a comparison between LP-based separation as implemented by [42] and the FW approach in the appendix. The FW approach is faster in run time but the gap closed is less than the LP-based approach. I am not sure whether it is enough to claim a clear advantage for the proposed algorithm. However, we should note that Table 3 shows promising results for the FW algorithm.


-	Since this paper does not propose a new theory and analysis, I don’t find the experimental evaluation comprehensive enough. The authors should have considered other problems, such as TSP, while including existing algorithms in their runs.

**Questions:**

-	How do you set $\epsilon$ in your algorithm?
-	Have you compared your algorithm numerically with other proposed methods for solving MKP or other popular IP problem (other than the results in the appendices)?
-	Have you run experiments for other problem instances?
-	I am curious about the possible shortcomings of the FW approach. The linear minimization oracle for the FW method uses dynamic programming, which introduces an overhead for the computation of $v_t$. Is there any regime where the FW approach with the proposed LMO oracle becomes computationally expensive such that the run times are comparable to LP-based approaches?

**Limitations:**

-	The authors might need to consider more problems like MKP to validate the performance of their approach.
-	I would prefer to see several other methods to be included in the numerical tests.

---

> ### Author Rebuttal · Authors · 2023-08-09
>
>
> 1. How do you set $\epsilon$ in your algorithm?
>
> Our algorithm sets $\epsilon$ to 1e-9, following the SCIP default. We will include this into the revised version of the paper.
>
> 2. Have you compared your algorithm numerically with other proposed methods for solving MKP or other popular IP problems (other than the results in the appendices)?
>
> Regarding the comparison of our algorithm with other methods, we have chosen to focus on a generic method capable of operating on any IP given a suitable lifting routine. We deemed it most relevant to test this within a general-purpose solver, rather than specialized heuristic methods. Our choice of SCIP is motivated by its status as the most widely used academic solver of this kind. The comparisons presented in the appendices involve the most recent state-of-the-art competing methods, as per our knowledge.
>
> 3. Have you run experiments for other problem instances?
>
> Concerning the testing of different problem instances, we used the MKP to introduce and illustrate our methodology, given its relevance and popularity in IP problems, thereby providing a robust benchmark. However, we plan to explore other problem classes in future work, with a particular interest in problems with trivial lifting, which we believe align well with our approach.
>
> 4. I am curious about the possible shortcomings of the FW approach. The linear minimization oracle for the FW method uses dynamic programming, which introduces an overhead for the computation of $v_t$. Is there any regime where the FW approach with the proposed linear optimization oracle becomes computationally expensive such that the run times are comparable to LP-based approaches?
>
> Please note that the oracle _used in the MKP_ uses dynamic programming. The structure of the oracle is very much problem-dependent, e.g., in the case of a TSP, the oracle could be enumerating a set of round tours on a subset of cities. Hence, one has to find a balance between the overhead introduced by calling the oracle and the advantages of the FW procedure in general.
>
> As for potential limitations of the FW approach, while it is technically possible to craft cases where FW runtimes exceed those of the LP-based approach, the overall effect on the solving process is more intricate. It depends on the quality of the cuts produced and their interplay with the various components of a modern MIP solver. This interaction encompasses aspects such as the cut filtering implementation, the strength of other obtained cuts, the solver's emphasis on generating cutting planes compared to other techniques, and more. Consequently, the method's true potential can only be gauged empirically over a relevant test set, as demonstrated in our computational studies by comparing out-of-the-box SCIP with SCIP enhanced by our method, as well as comparisons against SCIP enhanced with competing methods.

---

> > ### Comment · Area_Chair_Yeki · 2023-08-20
> >
> > Dear Reviewer,
> >
> > The author-reviewer discussion period is closing soon, so could you please go over the authors' rebuttal and respond with a message to the authors? It is important that authors receive a reply to their rebuttals, as they have tried to address comments raised by the reviewers.
> >
> > Best regards,
> > AC

---

> > ### Comment · Reviewer_cm37 · 2023-08-21
> > **Thank you for your response**
> >
> > I have read the responses by the authors regarding my questions.
> >
> > I understand that the reason behind the comparisons against SCIP rather than heuristics-based approaches. However, I still think it would complete the picture if the authors compare their methods against multiple existing algorithms and consider multiple IP problems to validate their claims with firm, numerical results. Nonetheless, I appreciate the alternative perspective to make use of FW-based sub-routine in IP solvers.
> >
> > I think this paper needs further improvements, as indicated in my initial comments, but I will increase my score after the authors response and reading the comments of the other reviewers.

---

> > > ### Author Response · Authors · 2023-08-22
> > >
> > > Thank you for your comment. We appreciate your perspective and we will revise the paper for the camera-ready version and broaden the computational results accordingly.

---

### Author Rebuttal · Authors · 2023-08-09

Dear Reviewers,

We sincerely thank you all for reviewing our paper. Your insights and suggestions have been incredibly valuable in refining our work and will undoubtedly lead to enhancing the quality and impact of our research. We appreciate your time and dedication to peer review and are grateful for your contributions.

Below you will find our answers to each specific question you raised. Additionally, we use the optional PDF page for figures and tables to provide the updated Figure 1, following the incorporation of comments from Reviewer kHh3.

Thank you once again for your valuable feedback.

Sincerely,

Authors

---

### Decision · Program_Chairs · 2023-09-21

**Decision:**

Accept (poster)

**Comment:**

The paper focuses on the cutting plane methods for integer programming and presents an efficient separation oracle that first attempts to find facets of a polytope in reduced dimensions (by solving the associated subproblem by using the Frank-Wolfe algorithm), and subsequently transforming them into cutting planes for the actual problem. While some reviewers raised concerns that the method falls out of the scope of the conference, I believe that the paper fits within the Optimization subject area. Not only have both cutting plane and Frank-Wolfe methods received attention within this community across various applications, but this community has also played a significant role in the theoretical development of such optimization methods in the past. The remaining comments for the paper were largely positive, although many reviewers noted some shortcomings in the numerical experiments. I strongly encourage the authors to carefully consider all feedback and address these points in the final version of the paper.